# SMYD3 Modulates the HGF/MET Signaling Pathway in Gastric Cancer

**DOI:** 10.3390/cells12202481

**Published:** 2023-10-18

**Authors:** Katia De Marco, Martina Lepore Signorile, Elisabetta Di Nicola, Paola Sanese, Candida Fasano, Giovanna Forte, Vittoria Disciglio, Antonino Pantaleo, Greta Varchi, Alberto Del Rio, Valentina Grossi, Cristiano Simone

**Affiliations:** 1Medical Genetics, National Institute for Gastroenterology—IRCCS “Saverio de Bellis” Research Hospital, 70013 Castellana Grotte, Italy; katia.demarco@irccsdebellis.it (K.D.M.); martina.lepore@irccsdebellis.it (M.L.S.); elisabetta.dinicola@irccsdebellis.it (E.D.N.); paola.sanese@irccsdebellis.it (P.S.); candida.fasano@irccsdebellis.it (C.F.); giovanna.forte@irccsdebellis.it (G.F.); vittoria.disciglio@irccsdebellis.it (V.D.); antonino.pantaleo@irccsdebellis.it (A.P.); 2Institute for Organic Synthesis and Photoreactivity (ISOF), National Research Council of Italy (CNR), 40129 Bologna, Italy; greta.varchi@isof.cnr.it (G.V.); alberto.delrio@isof.cnr.it (A.D.R.); 3Innovamol Consulting Srl, 41126 Modena, Italy; 4Medical Genetics, Department of Precision and Regenerative Medicine and Jonic Area (DiMePRe-J), University of Bari Aldo Moro, 70124 Bari, Italy

**Keywords:** MET, SMYD3, gastric cancer, stemness activity

## Abstract

Gastric cancer (GC) is the third most deadly cancer worldwide. Considerable efforts have been made to find targetable drivers in order to improve patient outcomes. MET is one of the most important factors involved in GC initiation and progression as it plays a major role in GC invasiveness and is related to cancer stemness. Unfortunately, treatment strategies targeting MET are still limited, with a proportion of patients responding to therapy but later developing resistance. Here, we showed that MET is a molecular partner of the SMYD3 methyltransferase in GC cells. Moreover, we found that SMYD3 pharmacological inhibition affects the HGF/MET downstream signaling pathway. Extensive cellular analyses in GC models indicated that EM127, a novel active site-selective covalent SMYD3 inhibitor, can be used as part of a synergistic approach with MET inhibitors in order to enhance the targeting of the HGF/MET pathway. Importantly, our data were confirmed in a 3D GC cell culture system, which was used as a surrogate to evaluate stemness characteristics. Our findings identify SMYD3 as a promising therapeutic target to impair the HGF/MET pathway for the treatment of GC.

## 1. Introduction

Gastric cancer (GC) is the third leading cause of cancer-related death, following lung and liver cancer, accounting for almost 800,000 deaths worldwide in 2020 [1,2]. The incidence and prevalence of GC vary geographically, affecting men more than women [2]. Since it is usually diagnosed at an advanced stage, the survival rate of affected patients is less than one year [3]. Although targeted strategies have brought new hope to antitumor therapy, chemotherapy currently remains the mainstay of GC treatment, and options for advanced GC with high heterogeneity are still limited [4]. Indeed, patient prognosis and treatment response depend not only on the tumor stage but also on the expression and tumor-specific alteration of signaling pathways [5].

Receptor tyrosine kinases (RTKs) play major roles in normal growth, development, and regeneration processes in mammals, and their dysregulation by gene amplification, chromosomal rearrangement, gene mutation, and transcriptional upregulation can result in various human malignancies, including GC [6]. Among RTKs, the c-MET (or MET) protein, which is encoded by the MET proto-oncogene, has been identified as a promising therapeutic target in GC [7,8,9]. The only known physiological ligand of MET is hepatocyte growth factor (HGF) [10]. MET plays an important role both in physiological and pathological processes; whereas its normal activation is essential for embryonic development and tissue repair, aberrant activation of its signaling cascade is involved in tumorigenesis, suppression of apoptosis, induction of cell proliferation, migration, angiogenesis, and metastasis [7]. In addition, MET supports the stem-like phenotype of cancer cells, affects the expression and activity of stem cell markers, and is associated with chemoresistance [11]. We recently showed that MET is a novel SMYD3 interactor, and this interaction was validated in colorectal cancer (CRC) cell lines [12]. SMYD3 methyltransferase has been found to be overexpressed in several types of cancers, including GC, and its oncogenic activity has been linked to proliferation, cell cycle regulation, increased migration, and invasion of cancer cells [13]. SMYD3 is an epigenetic regulator that methylates both histone and nonhistone proteins, orchestrating their interactions and functions [13]. 

Considering the oncogenic functions of MET and the emerging involvement of SMYD3 in GC, here we investigated the role of SMYD3 as a protein partner of MET in this cancer type and elucidated their structural and functional interplay. 

## 2. Materials and Methods

### 2.1. Cell Cultures

The AGS, NCI-N87, and KATO III GC cell lines were purchased from ATCC and cultured in RPMI high glucose without pyruvate (21875-034, Gibco, Carlsbad, CA, USA) with 10% FBS (10270-106, Gibco, Carlsbad, CA, USA) and 100 IU/mL penicillin-streptomycin (15140-122, Gibco, Carlsbad, CA, USA). The HGC-27 GC cell line, the HT29 CRC cell line, and the MCF7 breast cancer cell line were purchased from ATCC and cultured in DMEM high glucose without pyruvate (11360-070, Gibco, Carlsbad, CA, USA) with 10% FBS (10270-106, Gibco, Carlsbad, CA, USA) and 100 IU/mL penicillin–streptomycin (15140-122, Gibco, Carlsbad, CA, USA). Primary cancer-associated fibroblasts (CAFs) and normal fibroblasts (NFs) were isolated from gastric adenocarcinoma samples and normal gastric tissue, respectively, obtained from surgery. Samples were collected from a GC patient who signed an informed consent form in accordance with the ethical standards of the responsible Institutional Committee (Ethics Committee name: Comitato Etico Istituto tumori “Giovanni Paolo II”, Istituto di ricovero e cura a carattere scientifico, viale Orazio Flacco, 65-70124, Bari; approval code: Prot. n. 379/C.E.; approval date: 16 September 2020). Then, CAFs and NFs were incubated in IMDM high glucose (21980-032, Gibco, Carlsbad, CA, USA) with 20% FBS (10270-106, Gibco, Carlsbad, CA, USA) and 100 IU/mL penicillin–streptomycin (15140-122, Gibco, Carlsbad, CA, USA). All cell cultures were maintained in a humidified incubator at 37 °C and 5% CO_2_ and were tested to be mycoplasma free (#117048, Minerva Biolabs, Berlin, Germany) multiple times throughout the study.

To evaluate MET phosphoactivation, AGS and NCI-N87 cells were serum starved for 24 h and then treated with 50–300 ng/mL of HGF (#100-39, Peprotech, Waltham, MA, USA) for up to 1 h. 

### 2.2. MET and SMYD3 Inhibitors

For GC cell treatment, we used the MET inhibitor SU11274 (HY-12014, MedChemExpress, Monmouth Junction, NJ, USA) and the SMYD3 inhibitor EM127, which was synthesized as previously described in Parenti et al. [14]. Both inhibitors were dissolved in DMSO and stored at −80 °C. For each inhibitor, concentrations and treatment duration are indicated in the figure legends.

### 2.3. Three-Dimensional (3D) GC Cocultures and Tumorsphere Formation

To simulate in vivo conditions, 24-well ultralow attachment plates (3473, Corning, New York, NY, USA) were used for the 3D coculture of CAFs and GC cells. Cocultures were based on the combined culture of a 1:4 mix of GC cells and CAFs. The mixed suspensions were maintained in DMEM/F12 Advanced (12634010, Gibco, Carlsbad, CA, USA) supplemented with 6 mg/mL Glucose (G8769, Sigma-Aldrich, St. Louis, MO, USA), 2 mM L-Glutamine (25030081, Gibco, Carlsbad, CA, USA), 10 ng/mL bFGF (F0291, Sigma-Aldrich, St. Louis, MO, USA), 20 ng/mL EGF (E9644, Sigma-Aldrich, St. Louis, MO, USA), B27 supplement (17504044, Gibco, Carlsbad, CA, USA), and N-2 supplement (17502048, Gibco, Carlsbad, CA, USA). An inverted phase contrast microscope was used to observe the morphology and growth of developing 3D GC tumorspheres. 

### 2.4. Co-Immunoprecipitation Assays

Cells (4 × 10^6^ cells/100 mm dish) were treated or not with EM127 (5 μM) and SU11274 (10 μM) for 48 h, collected and homogenized in lysis buffer (50 mM Tris-HCl pH 7.4, 5 mM EDTA, 250 mM NaCl, and 1% Triton X-100) supplemented with protease and phosphatase inhibitors. Coupling between Dynabeads Protein A (10002D, Thermo Fisher Scientific, Waltham, MA, USA) and the selected antibodies, i.e., anti-SMYD3 (#12859, Cell Signaling Technology, Danvers, MA, USA), anti-MET (#8198, Cell Signaling Technology, Danvers, MA, USA), and anti-IgG (#2729 Cell Signaling Technology, Danvers, MA, USA), was performed in 100 μL of 0.01% Tween 20-1× PBS for 45 min at room temperature on a rocking platform. Then, samples were incubated with antibody–Dynabeads complexes for 3 h at room temperature on a rocking platform and immunoprecipitated. Immunoprecipitated proteins were extensively washed with lysis buffer, resuspended in Laemmli sample buffer, separated on a polyacrylamide gel, transferred to nitrocellulose membranes, and subjected to immunoblot analysis. Immunoblot analysis was performed using the following primary antibodies: anti-SMYD3 (#12859, Cell Signaling Technology, Danvers, MA, USA) and anti-MET (#8198, Cell Signaling Technology, Danvers, MA, USA). After incubation with rabbit IgG HRP (NA934V, GE Healthcare, Chicago, IL, USA) as a secondary antibody, the signal was revealed using the ECL plus chemiluminescence reagent (RPN2232, GE Healthcare, Chicago, IL, USA) according to the manufacturer’s instructions. Input corresponds to 10% of the whole cell lysate.

### 2.5. Immunoblotting

Whole cell extracts were obtained from cells collected and homogenized in lysis buffer (50 mM Tris-HCl pH 7.4, 5 mM EDTA, 250 mM NaCl, and 1% Triton X-100) supplemented with protease and phosphatase inhibitors. Between 20 and 40 μg of protein extracts from each sample was denatured in Laemmli sample buffer, loaded into an SDS-poly-acrylamide gel, and then transferred to nitrocellulose membranes for immunoblot analysis. Immunoblot analysis was performed using the following primary antibodies: anti-β-ACTIN (#3700, Cell Signaling Technology, Danvers, MA, USA), anti-SMYD3 (#12859, Cell Signaling Technology, Danvers, MA, USA), anti-phospho-p44/42 MAPK (ERK 1/2) (Thr202/tyr204) (#9106, Cell Signaling Technology, Danvers, MA, USA), anti-p44/42 MAPK (ERK 1/2) (#9102, Cell Signaling Technology, Danvers, MA, USA), anti-phospho-AKT (Ser473) (#9271, Cell Signaling Technology, Danvers, MA, USA), anti-AKT (#9272, Cell Signaling Technology, Danvers, MA, USA), anti-phospho-MET (Tyr1349) (#3121, Cell Signaling Technology, Danvers, MA, USA), anti-MET (#8198, Cell Signaling Technology, Danvers, MA, USA), anti-α-Smooth Muscle Actin (α-SMA) (A5228, Sigma-Aldrich, St. Louis, MO, USA), anti-Vimentin (#5741, Cell Signaling Technology, Danvers, MA, USA), anti-cleaved PARP (#5625, Cell Signaling Technology, Danvers, MA, USA), anti-cleaved Caspase 3 (#9661, Cell Signaling Technology, Danvers, MA, USA), and anti-GAPDH (#5174, Cell Signaling Technology, Danvers, MA, USA). After incubation with the secondary antibodies, i.e., rabbit IgG HRP and mouse IgG HRP (NA934V and NA931V, respectively, GE Healthcare, Chicago, IL, USA), the signal was revealed using the ECL-plus chemiluminescence reagent (RPN2232, GE Healthcare, Chicago, IL, USA) according to the manufacturer’s instructions. A densitometric evaluation was performed by Image Lab software (Bio-Rad Laboratories, Munchen, Germany) [15]. 

### 2.6. Cellular Assays

For cell viability assays, the CellTiter-Glo Luminescent Cell Viability Assay Kit (G7570, Promega, Madison, WI, USA) was used according to the manufacturer’s instructions. Briefly, one day before treatment, 5 × 10^3^ cells were seeded into 96-well plates using the PIPETMAX automatic platform (Gilson, Madison, WI, USA). Then, cells were treated or not with increasing concentrations of EM127 and/or SU11274 for up to 48 h. At the end of the treatment, 10 μL of the CellTiter-Glo Luminescent Cell Viability Reagent was added to each well and incubated at 37 °C in a humidified incubator for up to 1 h. The luminescent signal was read using a SPECTROstar Omega microplate reader (BMG Labtech, Ortenberg, Germany). 

Cell death was assessed by counting. Briefly, AGS and NCI-N87 GC cell lines were treated or not with EM127 (5 μM) and SU11274 (10 μM), as single or combined treatment, for 48 h. Then, supernatants (containing dead/floating cells) were collected. Cell pellets were resuspended in 1× PBS, and 10 μL was mixed with an equal volume of 0.01% Trypan blue solution (T8154, Sigma-Aldrich, St. Louis, MO, USA). Viable cells (unstained, trypan blue-negative cells) and dead cells (stained, trypan blue-positive cells) were counted with a phase-contrast microscope, and the percentages of viable and dead cells were calculated. 

Three-dimensional coculture tumorspheres were treated or not with EM127 (5 μM) and SU11274 (10 μM), as single or combined treatment, for 48 h, and then stained using the LIVE/DEAD^®^ Cell Imaging Kit (R37601, Thermo Fisher Scientific, Waltham, MA, USA) according to the manufacturer’s instructions. Specifically, live cells were stained in green (Calcein AM), and dead cells were stained in red (BOBO-3 Iodide). Digital image acquisition was carried out with a Zeiss Axio Observer fluorescence microscope using a 10× magnification objective. Quantification of cell death induction in tumorspheres was performed by analyzing the intensity of the red signal using ZEN blue software 3.3 version (Zeiss). The tumorsphere area (mm^2^) and roundness as 4 × area/(π × major_axis^2^) were calculated using an inbuilt feature of ImageJ software 5.2.1 version.

### 2.7. Colony Formation Assay

Colony formation assays were performed as previously described in Germani et al. [16]. Briefly, AGS and NCI-N87 GC cell lines were seeded as single cells in 12-well plates and allowed to attach for 2 h. Then, GC cells were treated with EM127 (5 μM) and/or SU11274 (10 μM). After 10 days of treatment, media were discarded, and cells were washed twice with 1× PBS. Cells were fixed with 4% paraformaldehyde (sc-281692, ChemCruz, Santa Cruz, CA, USA) for 20 min and then stained with Crystal violet solution (80299, Liofilchem, Roseto degli Abruzzi (TE), Italy). Then, cells were washed with water several times to remove excess Crystal violet solution, and plates were dried at room temperature.

### 2.8. RNA Isolation, cDNA Preparation, and qPCR Analysis

Total RNA of AGS and NCI-N87 GC cell lines treated or not with EM127 (5 μM) for 48 h was extracted using the PureLink RNA Micro Kit (12183-016, Invitrogen, Waltham, MA, USA) according to the manufacturer’s instructions. cDNA was synthesized on the RNA template (1 μg) using the SuperScript III Reverse Transcriptase (11755-050, Invitrogen, Waltham, MA, USA) according to the manufacturer’s instruction. qRT-PCR was performed in triplicate using the PowerUp SYBR Green Master Mix (A25741, Invitrogen, Waltham, MA, USA) on a QuantStudio Real-Time PCR System (Thermo Fisher Scientific, Waltham, MA, USA). qRT-PCR reactions were normalized using β-ACTIN as a housekeeping gene. Relative quantification was performed using the ddCT method. Primer sequences (β-ACTIN, CDK2, TERT, and WNT10B) are available upon request.

### 2.9. Quantification and Statistical Analysis

Data were analyzed and plotted using Microsoft Excel 2019, ImageJ 5.2.1 version, and ZEN blue microscopy software 3.3 version. The statistical significance of the results was analyzed using Student’s *t*-test, and *p* < 0.05 was considered statistically significant. Results are representative of at least three independent experiments.

## 3. Results

### 3.1. MET and SMYD3 Are Molecular Partners in GC Cells

Activation of MET is associated with many cancers, and several genes that are regulated by this signaling pathway are crucial for cancer initiation and progression [17,18]. In order to investigate the functional interaction between SMYD3 and MET in GC, we analyzed their expression levels in human GC cell lines. Our results showed that, among the GC cell lines tested, AGS, NCI-N87, and KATO-III express high levels of MET, while AGS, NCI-N87, and HGC-27 express high levels of SMYD3 (Figure 1a). MCF7 breast cancer cells were used as a control of low MET expression (https://maayanlab.cloud/Harmonizome/gene_set/MCF7/CCLE+Cell+Line+Gene+Mutation+Profiles, accessed on 3 July 2023) [19], while HT29 CRC cells were used as a control of SMYD3 overexpression (Figure 1a) [20,21].

We then used the AGS and NCI-N87 GC cell lines, both expressing high levels of MET and SMYD3, as cellular models to ascertain whether SMYD3 and MET are molecular partners in GC cells. Immunoprecipitation of whole cell lysates with an antiserum against SMYD3 or MET, followed by immunoblotting, indicated that SMYD3 interacts with MET in both GC cell lines (Figure 1b). 

### 3.2. Pharmacological Targeting of SMYD3 Abrogates the HGF/MET-Dependent Signaling Pathway

Since MET phosphorylation is required for the activation of its downstream pathway, and HGF is the only known physiological MET ligand [10], we evaluated the HGF dose–time response in AGS and NCI-N87 GC cellular models (Figure 2a). We found that treating serum-starved AGS and NCI-N87 cells with 300 ng/mL of HGF for 30 min induced significant phosphoactivation of MET (Figure 2a). Consistently, two different MET downstream targets, AKT and ERK 1/2, were phosphoactivated upon treatment of GC cells with 300 ng/mL of HGF for 30 min (Figure 2b). Of note, this treatment did not affect SMYD3 expression levels (Figure 2b).

In order to characterize the relationship between MET and SMYD3 in GC cells, we first tested the efficacy of a selective pyrrole–indolinone MET inhibitor (SU11274) in both GC cell lines. This compound has been reported to inhibit HGF-induced MET receptor phosphorylation in a dose-dependent manner [22]. To this end, GC cells were treated with SU11274 at increasing doses and times. Our results revealed that SU11274 has a half maximal inhibitory concentration (IC50) of 10 μM at 48 h in both AGS and NCI-N87 cell lines, as shown by the reduced proliferative index (Figure 3a). In these conditions, treatment with SU11274 also promoted a significant increase in cell death (Figure 3b). These data were further supported by immunoblot analyses showing that SU11274 inhibits the phosphoactivation of MET, ERK 1/2, and AKT (Figure 3c). As observed with HGF treatment, SU11274 treatment did not affect SMYD3 expression levels (Figure 3c). Altogether, these results indicate that treatment with 10 μM SU11274 for 48 h impairs MET activity in these GC cell lines.

To investigate more in depth the role of SMYD3 in MET signaling, we evaluated the efficacy of the novel active site-selective covalent SMYD3 inhibitor EM127 in our GC cell lines. This compound is a 4-aminopiperidine derivative bearing a 2-chloroethanoyl group as a reactive warhead that shows selectivity for the Cys186 residue located in the SMYD3 substrate/histone-binding pocket [14]. Therefore, GC cells were treated with EM127 at increasing doses and times. Our results showed that treatment with 5 μM EM127 for 48 h decreased AGS and NCI-N87 proliferative index (Figure 4a) and promoted cell death (Figure 4b). Intriguingly, in these conditions, EM127 also reduced the levels of phospho-MET, which in turn inhibited the phosphoactivation of AKT and ERK 1/2 (Figure 4c). Importantly, EM127 treatment did not affect SMYD3 expression levels (Figure 4c). Thus, to better characterize SMYD3 pharmacological inhibition by this compound, we treated or not AGS and NCI-N87 GC cell lines with 5 μM EM127 for 48 h and then analyzed the mRNA levels of known SMYD3 target genes by RT-qPCR. Our results showed that EM127 significantly reduced the expression of cyclin-dependent Kinase 2 (CDK2) (Appendix A), which is known to be regulated by SMYD3 [21,23]. Similar results were observed for WNT10B and TERT (Appendix A), two other SMYD3 downstream target genes [13]. These data indicate that EM127 specifically inhibits SMYD3 activity.

Altogether, these findings revealed that SMYD3 plays a crucial role in the HGF/MET signaling pathway. Indeed, they suggest that SMYD3 sustains MET oncogenic activity as its inhibition can reduce the active phosphorylated form of two key factors, AKT and ERK 1/2 kinases, which are both involved in cell proliferation, differentiation, migration, and death [24,25]. 

### 3.3. SMYD3 Inhibition Enhances GC Cell Sensitivity to MET Inhibition 

Since tyrosine kinase inhibitors targeting the HGF/MET pathway have already been studied in MET-positive GC with no substantial benefit [26], we evaluated the potential of EM127 as a sensitizing agent as part of a synergistic approach with SU11274. To this end, GC cell lines were treated with 5 μM EM127 and/or 10 μM SU11274 for 48 h. The efficacy of the combined treatment (EM127 and SU11274) in inhibiting the phosphoactivation of MET, ERK 1/2, and AKT was validated by immunoblotting analysis in both AGS and NCI-N87 GC cell lines (Appendix A). Moreover, our results revealed that this combined treatment is more effective than SU11274 alone. Indeed, concomitant use of EM127 and SU11274 further reduced the proliferative index and survival of GC cell lines compared with each treatment alone (Figure 5a,b) while promoting increased cell death (Figure 5c). Then, we carried out co-immunoprecipitation studies to ascertain whether the enzymatic activity of SMYD3 and MET is required for their physical interaction. Our data showed that SMYD3 and MET pharmacological inhibition by EM127 and SU11274, respectively, does not prevent the formation of an SMYD3 and MET complex (Appendix A). These data support the potential of SMYD3 inhibition to enhance the targeting of the HGF/MET pathway.

### 3.4. Targeting SMYD3 in 3D GC Cellular Models to Circumvent Stemness-Related MET Activity

It has been shown that the MET receptor is a critical factor responsible for the functional cancer stem cell (CSC) phenotype in various tumors, including GC [11]. In order to evaluate the role of MET in GC stemness, we established a 3D tumor-derived spheroid model that is enriched in CSCs or cells with stem cell-like properties [27]. Spheroids are grown as floating spheres and have been used to evaluate CSC-related characteristics of solid tumors in vitro [27]. 

To this end, AGS and NCI-N87 cells were cultured in ultralow attachment plates. In these conditions, the cells formed 3D floating clusters, which are commonly known as tumorspheres (Figure 6a). To explore the influence of HGF in this process, we treated these 3D cell cultures with 300 ng/mL of HGF for 48 h. As shown in Figure 6a, HGF enhanced their sphere formation ability, with an increase in sphere size and the acquisition of a distinctive morphology with a well-rounded shape (Appendix A). These results suggest that HGF influences the sphere structure.

Moreover, to better simulate the in vivo tumor microenvironment, we set up a 3D coculture system with GC cell lines and patient-derived CAFs. It is well known that cytokines secreted by CAFs play a significant role in tumor growth; among these cytokines, HGF is secreted mainly by CAFs and acts on MET-positive cancer cells in the tumor microenvironment [15]. Interactions between CAFs and cancer cells activate the HGF/MET signaling pathway, leading to tumor growth and metastasis [15]. 

Normal and cancer-associated primary fibroblasts were isolated from gastric peritumoral and adenocarcinoma samples, respectively, collected from a GC patient (Appendix A). CAFs were validated by analyzing the expression of the specific marker α-SMA by immunoblot analysis. Our results showed that α-SMA levels were higher in CAFs than in normal fibroblasts (NFs), confirming that our primary CAFs were activated fibroblasts (Appendix A). On the other hand, vimentin, a specific marker of stromal cells [14], was equally expressed in CAFs and paired NFs (Appendix A). 

The tumorspheres grown in 3D cocultures with CAFs were significantly larger than the tumorspheres grown as monocultures, and their morphology was characterized by a more rounded shape (Figure 6b and Appendix A). Since MET is involved in cancer stemness [11], GC tumorspheres grown in cocultures were treated with 10 μM SU11274 for 48 h. Our results showed that MET pharmacological inhibition affects tumorsphere viability (Figure 6b and Appendix A). Moreover, we evaluated the potential of EM127 as a sensitizing agent to enhance the effect of MET inhibition in 3D cocultures. To this end, tumorspheres grown as cocultures were treated with 10 μM SU11274 and/or 5 μM EM127 for 48 h and subjected to a live and dead staining assay. Our results showed a marked reduction in cell survival upon combined treatment (Figure 6b and Appendix A) compared with each treatment alone. Activation of the apoptotic pathway in cotreated GC tumorspheres was further confirmed by immunoblotting for cleaved PARP (PARP p85) and cleaved Caspase 3 (Figure 6c).

Overall, our data showed that the combined use of MET and SMYD3 inhibitors is a more efficient treatment strategy not only in GC cells grown as monolayer cultures but also in a 3D GC tumorsphere model. 

## 4. Discussion

The HGF/MET pathway has gained increasing interest in recent years for its involvement in cell growth, survival, epithelial–mesenchymal transition (EMT), metastasis, stemness, and chemoresistance in several tumor types [28]. In particular, the aberrant activation of MET signaling has been associated with GC progression and poor prognosis in GC patients [29]. 

Since effective treatment of GC remains a therapeutic challenge, recent studies have investigated the potential of targeted therapy [30,31]. In this context, MET represents an attractive molecular target in this tumor type. Indeed, preclinical studies suggested that MET inhibitors are promising agents in GC, and several MET inhibitors have been developed and tested in clinical trials [32,33]. 

However, the results showed limited clinical efficacy in GC patients due to a variety of reasons, including the development of resistance mechanisms, the intrinsic complexity of the MET pathway, and inadequate identification of relevant biomarkers [23]. 

In a previous study, we showed that SMYD3 interacts physically with various crucial players involved in cancer pathways. Among these, MET emerged as a novel SMYD3 interactor in CRC cell lines [12]. Therefore, here, we investigated the role of MET and its interaction with SMYD3 in GC cell lines. 

Various downstream signaling pathways are activated as a result of HGF–MET interaction, including ERK 1/2 and AKT cascades. The activation of these pathways leads, in turn, to various cellular responses, such as cell proliferation, survival, increased migration, and invasion [34]. Most MET tyrosine kinase inhibitors are small molecules that competitively antagonize occupancy of the intracellular ATP-binding site, preventing phosphorylation, tyrosine kinase activation, and downstream signaling effectors [35]. 

In the present study, SU11274, a specific MET inhibitor, effectively reduced the proliferation of AGS and NCI-N87 GC cell lines. In addition, phospho-MET levels were significantly decreased upon treatment with SU11274, resulting in the inhibition of ERK 1/2 and AKT activation. Moreover, the SMYD3 methyltransferase emerged as a molecular interactor of MET in GC cells, playing a key role in the HGF/MET signaling pathway. Indeed, our results showed that pharmacological inhibition of SMYD3 suppressed the growth and proliferation of AGS and NCI-N87 GC cells by downregulating AKT and ERK 1/2 phosphorylation. 

Cell culture systems are a powerful tool in cancer biology research [36]. For a long time, experiments have been performed by using two-dimensional (2D) cell cultures in vitro [36]. Unfortunately, these systems have some limitations, as they fail to reproduce important features of their in vivo counterparts, including cell polarity, morphology, and division [37]. These drawbacks led to the development of models that better recapitulate tumor characteristics, such as tumor-derived 3D spheroid cultures [36]. Indeed, spheroids are useful for studying novel anticancer strategies because they are enriched in CSCs or cells with stem cell-related characteristics [27,38]. Moreover, given the importance of the tumor microenvironment, cell coculture systems comprising GC cell lines and CAFs have been developed as a model to study gastric carcinogenesis [39]. 

Therefore, we evaluated the role of MET in a 3D system obtained by coculturing GC cell lines and patient-derived gastric CAFs and found that the HGF/MET signaling axis plays a significant role in sphere formation and tumor growth. In this 3D tumorsphere model, we evaluated the efficacy of combined SMYD3 and MET pharmacological inhibition, showing that it promotes a significant reduction in cell viability. 

Overall, these results indicate that SMYD3 pharmacological inhibition may represent a novel strategy to enhance the targeting of the HGF/MET pathway for the treatment of GC and warrant further investigation to ascertain its potential in preclinical and, hopefully, clinical settings.

The development of combined therapies for better clinical efficacy is proving a promising approach in cancer treatment [40]. This might well be the case for strategies targeting MET in GC [40]. Indeed, while MET inhibitors alone have shown limited efficacy in GC treatment, suggesting that MET-targeted monotherapy might not be an effective option, novel combined therapeutic strategies may provide more desirable outcomes [40]. Indeed, preclinical studies investigating the combined inhibition of MET and other pathways have shown significant therapeutic efficacy in various cancer types [41,42]. Similarly, our results suggest that SMYD3 is a promising target to overcome the therapeutic limitations of MET monotherapy in GC. In this light, further studies are needed to corroborate the efficacy of SMYD3 and MET combined inhibition and to identify which tumors are most likely to be responsive to SMYD3- and MET-targeted therapies. These will be instrumental in establishing future directions for validating this novel therapeutic strategy.

## Figures and Tables

**Figure 1 cells-12-02481-f001:**
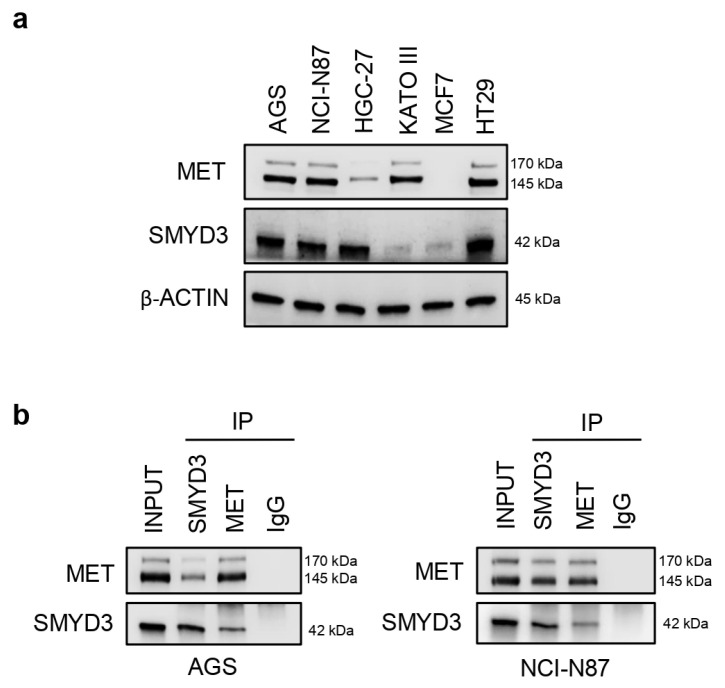
MET and SMYD3 are molecular partners in GC cells. (**a**) Immunoblot analysis showing MET and SMYD3 protein levels in various GC cell lines. The MCF7 breast cancer cell line and the HT29 CRC cell line were used as controls. (**b**) Co-immunoprecipitation of endogenous MET and SMYD3 in AGS and NCI-N87 GC cell lines. Input corresponds to 10% of the lysate. Anti-IgGs were used as negative controls.

**Figure 2 cells-12-02481-f002:**
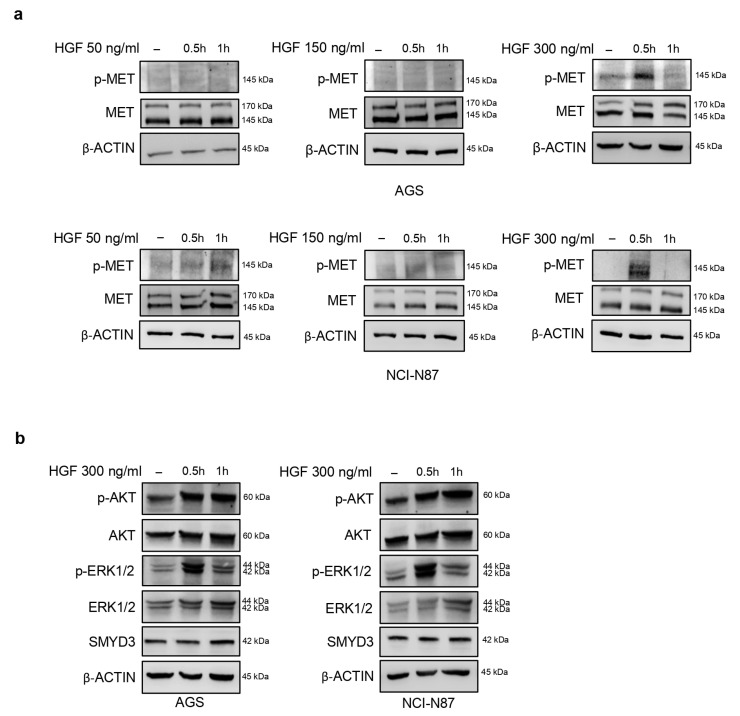
Activation of the MET pathway by HGF in GC cells. (**a**) Immunoblot analysis showing phosphoactivation of MET in response to increasing HGF doses and exposure time in GC cell lines. (**b**) Immunoblot analysis showing phosphoactivation of two MET targets (ERK 1/2 and AKT) in response to HGF in GC cell lines. β-ACTIN was used as a loading control.

**Figure 3 cells-12-02481-f003:**
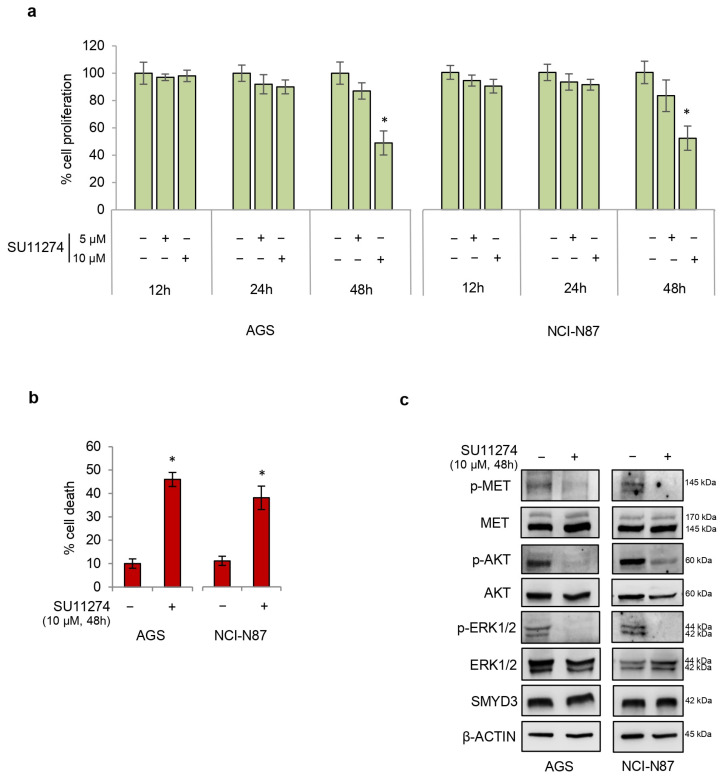
Treatment of GC cells with SU11274 decreases viability while increasing cell death. (**a**) Quantification of cell proliferation by Cell Titer Glo in GC cell lines. Cells were treated with increasing doses of SU11274 for up to 48 h. (**b**) Quantification of cell death by trypan blue staining in GC cell lines treated with 10 μM SU11274 for 48 h. (**c**) Immunoblot analysis showing phosphoactivation of MET, ERK 1/2, and AKT in GC cell lines treated with 10 μM SU11274 for 48 h. β-ACTIN was used as a loading control. * *p* < 0.05 treated vs. untreated.

**Figure 4 cells-12-02481-f004:**
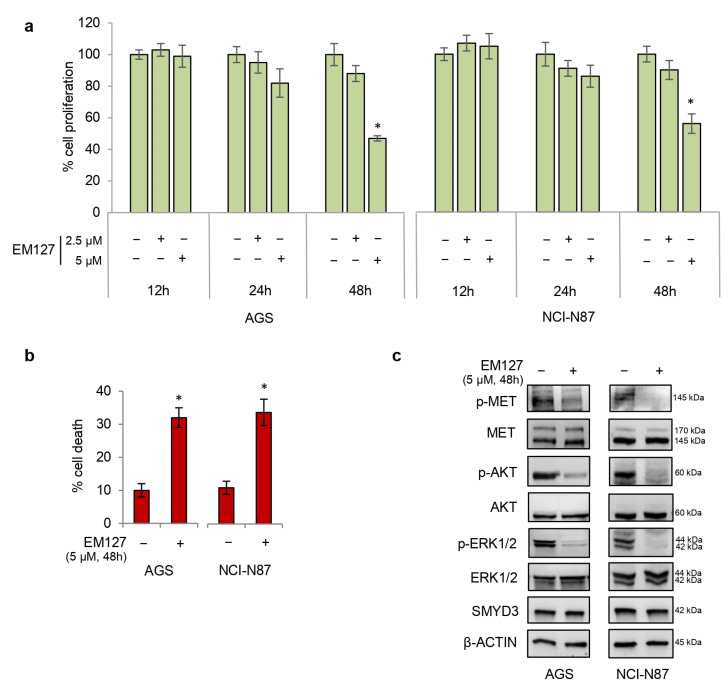
Treatment of GC cells with EM127 decreases viability while increasing cell death. (**a**) Quantification of cell proliferation by Cell Titer Glo in GC cell lines. Cells were treated with increasing doses of EM127 for up to 48 h. (**b**) Quantification of cell death by trypan blue staining in GC cell lines treated with 5 μM EM127 for 48 h. (**c**) Immunoblot analysis showing phosphoactivation of MET, ERK 1/2, and AKT in GC cell lines treated with 5 μM EM127 for 48 h. β-ACTIN was used as a loading control. * *p* < 0.05 treated vs. untreated.

**Figure 5 cells-12-02481-f005:**
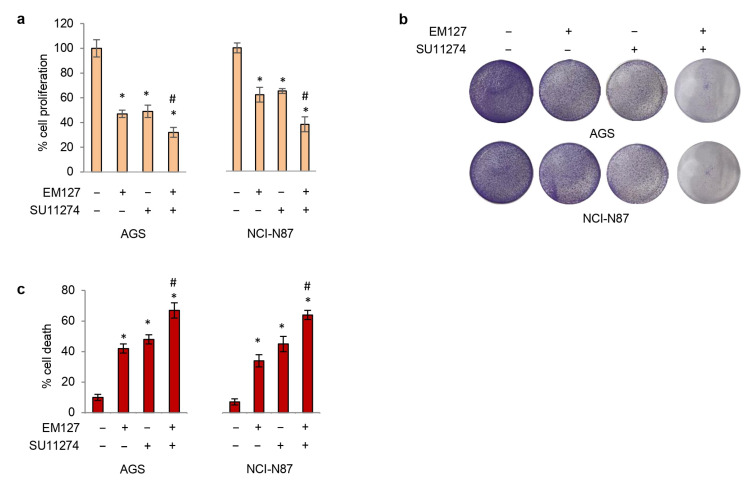
Targeting SMYD3 to enhance GC cell sensitivity to MET inhibition. (**a**) Quantification of cell proliferation by CellTiter-Glo in GC cell lines. Cells were treated with 5 μM EM127 and/or 10 μM SU11274 for 48 h. (**b**) Colony formation assay on GC cell lines treated as described in (**a**). (**c**) Quantification of cell death by trypan blue staining in GC cell lines treated as described in (**a**). * *p* < 0.05 treated vs. untreated. ^#^
*p* < 0.05 combined treatment vs. single treatments.

**Figure 6 cells-12-02481-f006:**
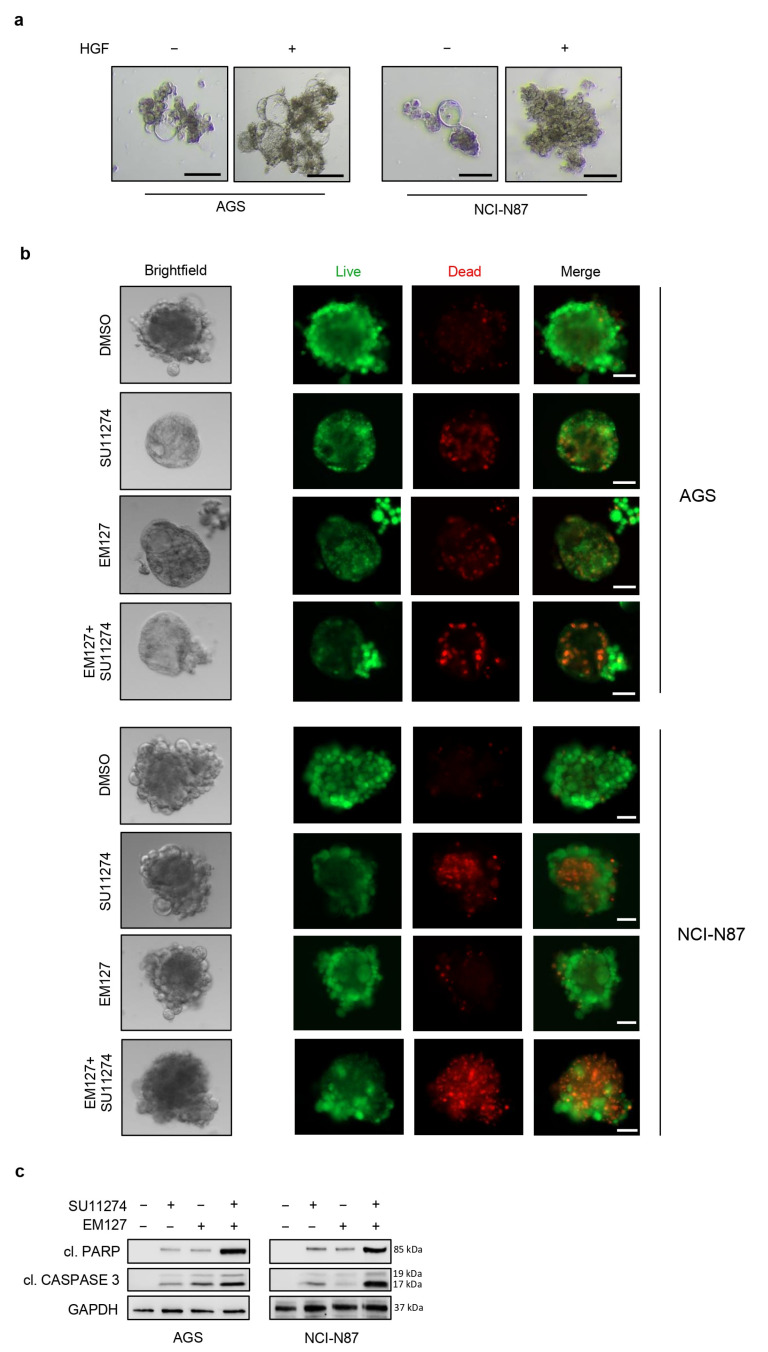
Combined treatment with EM127 and SU11274 has a synergistic cytotoxic effect in 3D GC cellular models. (**a**) Brightfield imaging of GC tumorspheres cultured with or without HGF (300 ng/mL). (**b**) Live and dead staining of GC tumorspheres grown as 3D cocultures with CAFs and treated with 5 μM EM127 and/or 10 μM SU11274 for 48 h. Green (Calcein AM): live cells; red (BOBO-3 Iodide): dead cells. Original magnification: 10×. (**c**) Immunoblot analysis showing cleaved PARP (p85) and cleaved Caspase 3 protein levels in GC tumorspheres grown and treated as described in (**b**). The scale bar represents 5 μm. GAPDH was used as a loading control. cl. PARP—cleaved PARP; cl. CASPASE 3—cleaved Caspase 3.

## Data Availability

Data sharing is not applicable to this article.

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
