# Peer review of "SMYD3 Modulates the HGF/MET Signaling Pathway in Gastric Cancer"

_cells, 2023, doi:10.3390/cells12202481_

Round 1

Reviewer 1 Report

In this manuscript the authors are addressing the potential for the inhibition of inhibition of MET receptor signaling in gastric cancer cells.

Firstly, the authors establish that their GC cell line models are expressing MET and examine whether HGF stimulation can activate MET signaling in these lines.  Once this was confirmed, they examined the effect of MET and SMYD3 in two GC lines.  Both showed IC50 effect on cell proliferation after 48h treatment.  Combining these two small molecules enhanced the cytotoxic effect.  Finally, spheroids were created from GC cell lines and incubated with patient derived CAFs.  Treatment of these tumourspheres with the inhibitors lead to cell death, especially upon co-treatment.

While this paper presents evidence for the importance of MET signaling in GC cell line survival, I am not adequately convinced of the potential of the inhibitors given that only cell lines and tumourspheres were utilised.  Where this paper would greatly benefit would be from the use of organoid models, or at the very least further interrogation of the tumoursphere model should be performed.  Quantification of the induction of cell death in the tumourspheres presented in figure 6c would be useful, as would additional examination of these tumourspheres after treatment- potentially some westerns or other signaling pathway analysis.   My primary concern with this manuscript is the heavy reliance on 2D models.

Other points to address:

-       The molecular weight of proteins should be marked on western figures.

-       In figure 1b the data for AGS cells is presented.  NCI-N87 data should also be shown here.

-       The westerns for p-MET are not of great quality.  Either the antibody is not optimised or of poor quality or the protein levels need to be increased, as the bands are diffuse and blurry in the majority of the blots.

-       Figure 6B should either be enlarged to make clearer or put in supplementary. It doesn’t really add to the manuscript.

-       An addition to the discussion regarding further avenues of investigation in this project could be useful.  

Author Response

Dear Editor,

we are pleased to submit the amended version of our work “SMYD3 modulates the HGF-MET signaling pathway in gastric cancer” (cells-2550389), which we would like to have considered for publication in Cells as part of the special issue “Non-coding RNA Regulation of Stem Cell Regenerative Mechanisms: Advances, Challenges, and Perspectives”. We addressed below all the reviewers’ comments by responding to their observations and by clarifying/adding sentences in the text in accordance with their suggestions.

Reviewer 1:

In this manuscript the authors are addressing the potential for the inhibition of inhibition of MET receptor signaling in gastric cancer cells.

Firstly, the authors establish that their GC cell line models are expressing MET and examine whether HGF stimulation can activate MET signaling in these lines. Once this was confirmed, they examined the effect of MET and SMYD3 in two GC lines. Both showed IC50 effect on cell proliferation after 48h treatment. Combining these two small molecules enhanced the cytotoxic effect. Finally, spheroids were created from GC cell lines and incubated with patient derived CAFs. Treatment of these tumourspheres with the inhibitors lead to cell death, especially upon co-treatment. While this paper presents evidence for the importance of MET signaling in GC cell line survival, I am not adequately convinced of the potential of the inhibitors given that only cell lines and tumourspheres were utilised. Where this paper would greatly benefit would be from the use of organoid models, or at the very least further interrogation of the tumoursphere model should be performed.

We thank the Reviewer for this comment. Below is a detailed point-by-point response to their remarks.

Quantification of the induction of cell death in the tumourspheres presented in figure 6c would be useful, as would additional examination of these tumourspheres after treatment- potentially some westerns or other signaling pathway analysis. My primary concern with this manuscript is the heavy reliance on 2D models.

We thank the Reviewer for these comments. In this amended version of the manuscript, we added the quantification of cell death induction in the tumorspheres shown in former Figure 6c (now Figure 6b). These data are now presented in new Supplementary Figure 2d. Moreover, we performed the suggested Western blotting experiments, in which we analyzed the levels of cleaved PARP (PARP p85) and cleaved Caspase 3 in both GC tumorsphere cultures used in this study. These results are now presented in new Figure 6c. In addition to these findings, we added new controls and analyses of our 3D models, which are now presented in new Supplementary Figure 2a.

Other points to address:

- The molecular weight of proteins should be marked on western figures.

As suggested by the Reviewer, in this amended version of the manuscript, we added the molecular weight of all proteins in each Western blot figure.

- In figure 1b the data for AGS cells is presented. NCI-N87 data should also be shown here.

We are grateful to the Reviewer for raising this point. As suggested, in this amended version of the manuscript, we included the co-immunoprecipitation experiments performed in NCI-N87 cells. These results are now presented in new Figure 1b.

- The westerns for p-MET are not of great quality. Either the antibody is not optimised or of poor quality or

the protein levels need to be increased, as the bands are diffuse and blurry in the majority of the blots.

We agree that in most Western blots for p-MET (Met (D1C2) XP® Rabbit mAb #8198, Cell Signaling Technology) the bands are not well defined, which is due at least in part to the low levels of p-MET in the culture conditions under evaluation. In this amended version of the manuscript, we improved the quality of the images as much as possible in order to more clearly show that MET is significantly phospho-activated after 30 minutes of treatment with 300 ng/ml HGF (see Figure 2a).

- Figure 6B should either be enlarged to make clearer or put in supplementary. It doesn’t really add to the

manuscript.

We are sorry about the quality of the figures accompanying the previous version of our manuscript. In our previous submission, we separately uploaded PDF files with high-resolution figures in accordance with Cells author guidelines, but something might have gone wrong. We have now confirmed that high-resolution figures have been correctly uploaded with this amended version of the manuscript and, as suggested by the Reviewer, we moved the schematic representation of the generation of 3D co-cultures in new Supplementary Figure 2b.

- An addition to the discussion regarding further avenues of investigation in this project could be useful.

We thank the Reviewer for this comment. As suggested, in this amended version of the manuscript, we outlined the future directions of our research at the end of the Discussion section.

Reviewer 2 Report

This article suggested through various experiments that SMYD3 regulates HGF-MET signaling in gastric cancer. However, the basic parts such as figure quality arrangement of the overall figure data are insufficient, and there are data that reduce reliability of some data. In particular, when SMYD3 expression was regulated after HGF treatment in various cancer cell lines including gastric cancer cell lines in 2009, MET expression was regulated and motility was regulated. Some are difficult to do (J.-N. Zou et al. / Cancer Letters 280 (2009) 78–85). In addition, there are several questions below, which suggest related questions.

1. IP was performed using SMYD3 antibody to confirm binding with MET, but on the contrary, IP data using MET seems to be additionally needed.

2. When confirmed using SU11274 and EM127 drugs, it seems necessary to verify with experiments how MET and SMYD3 interactions change.

3. In Figure 2, it is thought that it would be good to show the unity of time through the HGF treated time notation 30' -> 0.5 h correction.

4. Figure 2B. HGF-cMET signal can be seen as representative MAPK signals (JNK, p38, ERK), STAT, PI3K-AKT, and NF-kB. Among them, why were PI3K-AKT and ERK identified?

5. When performing the experiments in Figure 2 and 3, SMYD3 expression was not confirmed, but if SMUD3 is expected to be the top factor, SMYD3 expression was shown in the above experiment to confirm that no change was seen.

6. Figure 4A. Among % cell proliferation, the NCI-N87 control group seems to need to organize the percent of other groups based on 100%. The starting point seems ambiguous, leading to misunderstandings.

7. Figure 4C. SMYD3 inhibitor EM127 was used, but data on SMYD3 inhibition by SMYD3 inhibitor seems to be needed.

8. Figure 5. MET and SMYD3 inhibitor were co-treated, and MET and SMYD3 inhibition data by the above drug should be additionally attached. The reliability of the data may be reduced because the results for the inhibitor are not presented.

9. Figure 6A. The scale bar mark is missing, so it seems to need correction.

10. Figure 6A. It was mentioned that cell morphology increases and rounds during HGF treatment, but it is judged that there is no information such as the size, number, and rounding state of spheres formed during SMYD3 and MET inhibitor treatment. It seems that additional statistical data are needed.

11. Figure 6B. It is a high-definition picture and it seems to need correction.

12. Figure 6C. What reagents are used for green and red, and information about distinguishing between live and dead seems to be needed to stain which part.

13. Figure 6C. It seems necessary to additionally present an experiment to compare and confirm the 3D condition and the 3D+CAF condition.

14. IRB number missing, it seems to need correction.

Author Response

Dear Editor,

we are pleased to submit the amended version of our work “SMYD3 modulates the HGF-MET signaling pathway in gastric cancer” (cells-2550389), which we would like to have considered for publication in Cells as part of the special issue “Non-coding RNA Regulation of Stem Cell Regenerative Mechanisms: Advances, Challenges, and Perspectives”. We addressed below all the reviewers’ comments by responding to their observations and by clarifying/adding sentences in the text in accordance with their suggestions.

Reviewer 2:

This article suggested through various experiments that SMYD3 regulates HGF-MET signaling in gastric cancer. However, the basic parts such as figure quality arrangement of the overall figure data are insufficient, and there are data that reduce reliability of some data. In particular, when SMYD3 expression was regulated after HGF treatment in various cancer cell lines including gastric cancer cell lines in 2009, MET expression was regulated and motility was regulated. Some are difficult to do (J.-N. Zou et al. / Cancer Letters 280

(2009) 78–85).

We thank the Reviewer for this comment. Below is a detailed point-by-point response to their remarks.

In addition, there are several questions below, which suggest related questions.

  1. IP was performed using SMYD3 antibody to confirm binding with MET, but on the contrary, IP data using MET seems to be additionally needed.

We thank the Reviewer for this comment. In this amended version of the manuscript, we included co-immunoprecipitation experiments in AGS and NCI-N87 cells. Immunoprecipitations were carried out with an antiserum against SMYD3 or MET, followed by immunoblotting for SMYD3 and MET. We included these results in new Figure 1b.

  1. When confirmed using SU11274 and EM127 drugs, it seems necessary to verify with experiments how

MET and SMYD3 interactions change.

We are grateful to the Reviewer for this suggestion. In this amended version of the manuscript, we included the suggested experiment, whose results are now presented in Supplementary Figure 1c. Briefly, GC cells were treated with SU11274 (10 μM) and EM127 (5 μM) for 48 hours and then collected for co-immunoprecipitation experiments. Immunoprecipitations were carried out with an antiserum against SMYD3, followed by immunoblotting for SMYD3 and MET. Our results showed that pharmacological inhibition of MET and SMYD3 with SU11274 and EM127, respectively, does not prevent their interaction.

  1. In Figure 2, it is thought that it would be good to show the unity of time through the HGF treated time

notation 30' -> 0.5 h correction.

We are grateful to the Reviewer for this suggestion. In this amended version of the manuscript, we amended the notation of the HGF treatment time.

  1. Figure 2B. HGF-cMET signal can be seen as representative MAPK signals (JNK, p38, ERK), STAT, PI3KAKT, and NF-kB. Among them, why were PI3K-AKT and ERK identified?

We are grateful to the Reviewer for this comment. The PI3K/AKT/mTOR and MAPK/ERK pathways are the two major downstream effectors in MET-mediated signal transduction. We chose these two important kinases because AKT and ERK are two oncogenic drivers in GC; indeed, ERK 1/2 is a critical player involved in tumor proliferation, invasion, and metastasis, and AKT is involved in cell growth, survival, genome stability and neovascularization (Magnelli L, Schiavone N, Staderini F, Biagioni A, Papucci L, MAP Kinases Pathways in Gastric Cancer, Int. J. Mol. Sci. 2020, 21(8), 2893; doi.org/10.3390/ijms21082893. Matsuoka T, Yashiro M, The Role of PI3K/Akt/mTOR Signaling in Gastric Carcinoma, Cancers 2014, 6, 1441-1463; doi:10.3390/cancers6031441).

The significant role played by the PI3K/AKT/mTOR and MEK/ERK pathways upon activation by MET in the initiation and development of gastric carcinoma suggests that they may be appropriate targets to investigate in this cancer model.

  1. When performing the experiments in Figure 2 and 3, SMYD3 expression was not confirmed, but if SMYD3 is expected to be the top factor, SMYD3 expression was shown in the above experiment to confirm that no change was seen.

We are grateful to the Reviewer for this comment. In this amended version of the manuscript, we included the suggested experiments in both GC cell lines used in this study. These results are now presented in new Figures 2 and 3.

  1. Figure 4A. Among % cell proliferation, the NCI-N87 control group seems to need to organize the percent

of other groups based on 100%. The starting point seems ambiguous, leading to misunderstandings.

We thank the Reviewer for this suggestion. In this amended version of the manuscript, we corrected the graphs of the NCI-N87 control group in Figure 4a.

  1. Figure 4C. SMYD3 inhibitor EM127 was used, but data on SMYD3 inhibition by SMYD3 inhibitor seems to be needed.

We thank the Reviewer for this comment, as it allowed us to better describe an important point of our work. As suggested by the Reviewer, we evaluated the expression of SMYD3 upon its pharmacological inhibition and presented these results in new Figure 4c. EM127 did not change the expression of SMYD3 but affected its enzymatic activity. EM127 is a 4-aminopiperidine derivative bearing a 2-chloroethanoyl group as a reactive warhead that shows selectivity for the Cys186 residue located in SMYD3 substrate/histone binding pocket (Parenti MD, Naldi M, Manoni E, Fabini E, Cederfelt D, Talibov VO, Gressani V, Guven U, Grossi V, Fasano C, Sanese P, De Marco K, Shtil AA, Kurkin AV, Altieri A, Danielson UH, Caretti G, Simone C, Varchi G, Bartolini M, Del Rio A. Discovery of the 4-aminopiperidine-based compound EM127 for the site-specific covalent inhibition of SMYD3. Eur J Med Chem. 2022 Dec 5;243:114683. doi: 10.1016/j.ejmech.2022.114683. Epub 2022 Sep 8. PMID: 36116234.). In this light, to better characterize SMYD3 pharmacological inhibition, we treated AGS and NCI-N87 GC cells with EM127 and analyzed by RT-qPCR the mRNA levels of SMYD3 known target genes such as Cyclin-Dependent Kinase 2 (CDK2), WNT10B, and TERT. As expected, the expression levels of these genes decreased after treatment with EM127. These data are presented in new Supplementary Figure 1a.

  1. Figure 5. MET and SMYD3 inhibitor were co-treated, and MET and SMYD3 inhibition data by the above

drug should be additionally attached. The reliability of the data may be reduced because the results for the

inhibitor are not presented.

We thank the Reviewer for this suggestion. In this amended version of the manuscript, we evaluated the efficacy of the combined treatment (EM127+SU11274) by immunoblotting analysis and added these results in new Supplementary Figure 1b

  1. Figure 6A. The scale bar mark is missing, so it seems to need correction.

We are grateful to the Reviewer for this remark. In this amended version of the manuscript, we added the suggested scale bar marks in Figure 6a.

  1. Figure 6A. It was mentioned that cell morphology increases and rounds during HGF treatment, but it is judged that there is no information such as the size, number, and rounding state of spheres formed during SMYD3 and MET inhibitor treatment. It seems that additional statistical data are needed.

We thank the Reviewer for this comment. As suggested, in this amended version of the manuscript, we quantified tumorsphere size and roundness and presented these data as bar graphs in new Supplementary Figure 2a (left and right panel). The specifications of these findings are now described in the Materials and Methods section.

  1. Figure 6B. It is a high-definition picture and it seems to need correction.

We are sorry about the quality of the figures accompanying the previous version of our manuscript. In our previous submission, we separately uploaded PDF files with high-resolution figures in accordance with Cells author guidelines, but something might have gone wrong. We have now confirmed that high-resolution figures have been correctly uploaded with this amended version of the manuscript and, as suggested by the Reviewer, we moved the image in Supplementary Figure 2b.

  1. Figure 6C. What reagents are used for green and red, and information about distinguishing between live

and dead seems to be needed to stain which part.

We thank the Reviewer for this suggestion. In this amended version of the manuscript, we added this information in the figure legend. Specifically, the cell-permeant dye Calcein AM was used as a live cell indicator and BOBO-3 Iodide was used as a dead cell indicator.

  1. Figure 6C. It seems necessary to additionally present an experiment to compare and confirm the 3D condition and the 3D+CAF condition.

We thank the Reviewer for this comment. As stated above, in this amended version of the manuscript, we compared the 3D condition and the 3D+CAF condition by quantifying tumorsphere size and roundness and presented these data as bar graphs in new Supplementary Figure 2a (left and right panel).

  1. IRB number missing, it seems to need correction.

We thank the Reviewer for this suggestion. In this amended version of the manuscript, we added the following information in the Materials and Methods section: Ethics Committee name: Comitato Etico Istituto tumori "Giovanni Paolo II", Istituto di ricovero e cura a carattere scientifico, viale Orazio Flacco, 65-70124, Bari. Approval Code: Prot. n. 379/C.E. Approval Date: 16/09/2020.

Reviewer 3 Report

De Marco et al. described the new role of SMYD3 protein in GC cell growth and in HGF/MET signaling pathway and a new strategy to target GC cell growth. They investigated the role of SMYD3, a protein interactor of MET (a receptor tyrosine kinase) and a ligand of HGF, which is believed to be a promising target for GC treatment, due to its relevant role in GC aggressiveness. Since the exclusive targeting of MET is not an efficacious anticancer strategy for GC, the authors showed the efficacy of SMYD3 targeting, through a new pharmacological compound, EM127, able to block SMYD3 methyltransferase activity. Finally, by using 3D cellular models, they showed significant inhibitory effect on GC growth with combination treatment of a well-established MET inhibitor SU11274 and EM127.

The manuscript is novel and of broad interest. It is well written, reading is fluent, and concepts are clearly described. The following points are suggested:

1)    The drugs used in GC cellular models SU11274 and EM127 both induce apoptosis. If possible, this effect could be further demonstrated showing Cleaved CASPASE-3 and Cleaved PARP, or through ANNEXIN V/PI double staining.

2)    Figure 3 and 4: is not clear the concentration used and time of treatments in panel C of both figures. Could the authors specify in the figure legends? For the same panels: is the experiment performed upon HGF treatment or not?

3)    Figure 6C: Could authors add more technical details for the experiment performed?

Author Response

Dear Editor,

we are pleased to submit the amended version of our work “SMYD3 modulates the HGF-MET signaling pathway in gastric cancer” (cells-2550389), which we would like to have considered for publication in Cells as part of the special issue “Non-coding RNA Regulation of Stem Cell Regenerative Mechanisms: Advances, Challenges, and Perspectives”. We addressed below all the reviewers’ comments by responding to their observations and by clarifying/adding sentences in the text in accordance with their suggestions.

Reviewer 3:

De Marco et al. described the new role of SMYD3 protein in GC cell growth and in HGF/MET signalingpathway and a new strategy to target GC cell growth. They investigated the role of SMYD3, a protein interactor of MET (a receptor tyrosine kinase) and a ligand of HGF, which is believed to be a promising target for GC treatment, due to its relevant role in GC aggressiveness. Since the exclusive targeting of MET is not an efficacious anticancer strategy for GC, the authors showed the efficacy of SMYD3 targeting, through a new pharmacological compound, EM127, able to block SMYD3 methyltransferase activity. Finally, by using 3D cellular models, they showed significant inhibitory effect on GC growth with combination treatment of a well-established MET inhibitor SU11274 and EM127.

The manuscript is novel and of broad interest. It is well written, reading is fluent, and concepts are clearly

described.

We thank the Reviewer for these positive general comments. Below is a detailed point-by-point response to their remarks.

The following points are suggested:

1) The drugs used in GC cellular models SU11274 and EM127 both induce apoptosis. If possible, this effect

could be further demonstrated showing Cleaved CASPASE-3 and Cleaved PARP, or through ANNEXIN V/PI double staining.

We thank the Reviewer for this suggestion. In this amended version of the manuscript, we included the suggested Western blotting experiments showing the levels of both cleaved PARP (PARP p85) and cleaved Caspase 3 in the GC tumorsphere cultures used in this study. These results are now presented in new Figure 6c.

2) Figure 3 and 4: is not clear the concentration used and time of treatments in panel C of both figures. Could the authors specify in the figure legends? For the same panels: is the experiment performed upon HGF treatment or not?

We thank the Reviewer for this comment. As suggested, in this amended version of the manuscript, we added the concentration of the inhibitors and the duration of the treatments in the figure legends of both Figure 3c (SU11274 10μM for 48 hours) and 4c (EM127 5μM for 48 hours). These experiments were not performed upon HGF treatment.

3) Figure 6C: Could authors add more technical details for the experiment performed?

We thank the Reviewer for this comment. As suggested, in this amended version of the manuscript, we added further technical details about these experiments in the Materials and Methods section.

Round 2

Reviewer 1 Report

The quality of this manuscript has been significantly improved by the addition of a number of results. I would now recommend to accept the article.

Author Response

We thank the reviewers 1 for accepting the amended version of our manuscript.

Reviewer 2 Report

Although the authors appear to have solved many existing problems through their review, it is confirmed that there are still many problems. It seems that the following problems need to be resolved.

1. Although the title says that SMYD3 regulates HGF-MET signaling, this paper mainly focuses on inhibition of SMYD3. In order to show the mechanism of HGF-MET signal transduction by SMYD3, it is necessary to prove the mechanism of MET by SMYD3.

2. It is believed that the p-MET data shown in Figure 2a should be shown in one gel for concentration experiments of 50ng, 150ng and 300ng so that comparison of concentrations can be made clearer.

3. Figures 3C and 4C basically show that there is phosphorylation of MET, but looking at the data in Figure 2a, it appears that phosphorylation of MET is not high. I think a comparative explanation as to why they are different is needed. Did you process the inhibitor after 300ng of HGF?

4. I think the data in figure 6C may also be needed in figure 5c. Additionally, if you want to see cleaved caspase3 and PARP, it seems necessary to show the total form as well for comparison.

5. I think a subtitle for supplementary figures 1 and 2 is needed.

6. Although many corrections have been made, typos still exist.

ex) figure S21 -> figure S2

Author Response

Dear Editor,

we are pleased to submit the rebuttal letter of our work “SMYD3 modulates the HGF-MET signaling pathway in gastric cancer” (cells-2550389), which we would like to have considered for publication in Cells as part of the special issue “Non-coding RNA Regulation of Stem Cell Regenerative Mechanisms: Advances, Challenges, and Perspectives”. We addressed below the reviewer’ comments by responding to his/her observations in a detailed point-by-point response to his/her remarks.

Reviewer 1 and 3

We thank the reviewers 1 and 3 for accepting the amended version of our manuscript.

Reviewer 2

In order to reply to the points raised by the reviewer 2 in the second round of revision it is necessary to resume its first round of revision.

Review report (round 1) completed

This article suggested through various experiments that SMYD3 regulates HGF-MET signaling in gastric cancer. However, the basic parts such as figure quality arrangement of the overall figure data are insufficient, and there are data that reduce reliability of some data. In particular, when SMYD3 expression was regulated after HGF treatment in various cancer cell lines including gastric cancer cell lines in 2009, MET expression was regulated and motility was regulated. Some are difficult to do (J.-N. Zou et al. / Cancer Letters 280

(2009) 78–85).

We thank the Reviewer for this comment. Below is a detailed point-by-point response to their remarks.

In addition, there are several questions below, which suggest related questions.

  1. IP was performed using SMYD3 antibody to confirm binding with MET, but on the contrary, IP data using MET seems to be additionally needed.

We thank the Reviewer for this comment. In this amended version of the manuscript, we included co-immunoprecipitation experiments in AGS and NCI-N87 cells. Immunoprecipitations were carried out with an antiserum against SMYD3 or MET, followed by immunoblotting for SMYD3 and MET. We included these results in new Figure 1b.

  1. When confirmed using SU11274 and EM127 drugs, it seems necessary to verify with experiments how

MET and SMYD3 interactions change.

We are grateful to the Reviewer for this suggestion. In this amended version of the manuscript, we included the suggested experiment, whose results are now presented in Supplementary Figure 1c. Briefly, GC cells were treated with SU11274 (10 μM) and EM127 (5 μM) for 48 hours and then collected for co-immunoprecipitation experiments. Immunoprecipitations were carried out with an antiserum against SMYD3, followed by immunoblotting for SMYD3 and MET. Our results showed that pharmacological inhibition of MET and SMYD3 with SU11274 and EM127, respectively, does not prevent their interaction.

  1. In Figure 2, it is thought that it would be good to show the unity of time through the HGF treated time

notation 30' -> 0.5 h correction.

We are grateful to the Reviewer for this suggestion. In this amended version of the manuscript, we amended the notation of the HGF treatment time.

  1. Figure 2B. HGF-cMET signal can be seen as representative MAPK signals (JNK, p38, ERK), STAT, PI3KAKT, and NF-kB. Among them, why were PI3K-AKT and ERK identified?

We are grateful to the Reviewer for this comment. The PI3K/AKT/mTOR and MAPK/ERK pathways are the two major downstream effectors in MET-mediated signal transduction. We chose these two important kinases because AKT and ERK are two oncogenic drivers in GC; indeed, ERK 1/2 is a critical player involved in tumor proliferation, invasion, and metastasis, and AKT is involved in cell growth, survival, genome stability and neovascularization (Magnelli L, Schiavone N, Staderini F, Biagioni A, Papucci L, MAP Kinases Pathways in Gastric Cancer, Int. J. Mol. Sci. 2020, 21(8), 2893; doi.org/10.3390/ijms21082893. Matsuoka T, Yashiro M, The Role of PI3K/Akt/mTOR Signaling in Gastric Carcinoma, Cancers 2014, 6, 1441-1463; doi:10.3390/cancers6031441).

The significant role played by the PI3K/AKT/mTOR and MEK/ERK pathways upon activation by MET in the initiation and development of gastric carcinoma suggests that they may be appropriate targets to investigate in this cancer model.

  1. When performing the experiments in Figure 2 and 3, SMYD3 expression was not confirmed, but if SMYD3 is expected to be the top factor, SMYD3 expression was shown in the above experiment to confirm that no change was seen.

We are grateful to the Reviewer for this comment. In this amended version of the manuscript, we included the suggested experiments in both GC cell lines used in this study. These results are now presented in new Figures 2 and 3.

  1. Figure 4A. Among % cell proliferation, the NCI-N87 control group seems to need to organize the percent

of other groups based on 100%. The starting point seems ambiguous, leading to misunderstandings.

We thank the Reviewer for this suggestion. In this amended version of the manuscript, we corrected the graphs of the NCI-N87 control group in Figure 4a.

  1. Figure 4C. SMYD3 inhibitor EM127 was used, but data on SMYD3 inhibition by SMYD3 inhibitor seems to be needed.

We thank the Reviewer for this comment, as it allowed us to better describe an important point of our work. As suggested by the Reviewer, we evaluated the expression of SMYD3 upon its pharmacological inhibition and presented these results in new Figure 4c. EM127 did not change the expression of SMYD3 but affected its enzymatic activity. EM127 is a 4-aminopiperidine derivative bearing a 2-chloroethanoyl group as a reactive warhead that shows selectivity for the Cys186 residue located in SMYD3 substrate/histone binding pocket (Parenti MD, Naldi M, Manoni E, Fabini E, Cederfelt D, Talibov VO, Gressani V, Guven U, Grossi V, Fasano C, Sanese P, De Marco K, Shtil AA, Kurkin AV, Altieri A, Danielson UH, Caretti G, Simone C, Varchi G, Bartolini M, Del Rio A. Discovery of the 4-aminopiperidine-based compound EM127 for the site-specific covalent inhibition of SMYD3. Eur J Med Chem. 2022 Dec 5;243:114683. doi: 10.1016/j.ejmech.2022.114683. Epub 2022 Sep 8. PMID: 36116234.). In this light, to better characterize SMYD3 pharmacological inhibition, we treated AGS and NCI-N87 GC cells with EM127 and analyzed by RT-qPCR the mRNA levels of SMYD3 known target genes such as Cyclin-Dependent Kinase 2 (CDK2), WNT10B, and TERT. As expected, the expression levels of these genes decreased after treatment with EM127. These data are presented in new Supplementary Figure 1a.

  1. Figure 5. MET and SMYD3 inhibitor were co-treated, and MET and SMYD3 inhibition data by the above

drug should be additionally attached. The reliability of the data may be reduced because the results for the

inhibitor are not presented.

We thank the Reviewer for this suggestion. In this amended version of the manuscript, we evaluated the efficacy of the combined treatment (EM127+SU11274) by immunoblotting analysis and added these results in new Supplementary Figure 1b

  1. Figure 6A. The scale bar mark is missing, so it seems to need correction.

We are grateful to the Reviewer for this remark. In this amended version of the manuscript, we added the suggested scale bar marks in Figure 6a.

  1. Figure 6A. It was mentioned that cell morphology increases and rounds during HGF treatment, but it is judged that there is no information such as the size, number, and rounding state of spheres formed during SMYD3 and MET inhibitor treatment. It seems that additional statistical data are needed.

We thank the Reviewer for this comment. As suggested, in this amended version of the manuscript, we quantified tumorsphere size and roundness and presented these data as bar graphs in new Supplementary Figure 2a (left and right panel). The specifications of these findings are now described in the Materials and Methods section.

  1. Figure 6B. It is a high-definition picture and it seems to need correction.

We are sorry about the quality of the figures accompanying the previous version of our manuscript. In our previous submission, we separately uploaded PDF files with high-resolution figures in accordance with Cells author guidelines, but something might have gone wrong. We have now confirmed that high-resolution figures have been correctly uploaded with this amended version of the manuscript and, as suggested by the Reviewer, we moved the image in Supplementary Figure 2b.

  1. Figure 6C. What reagents are used for green and red, and information about distinguishing between live

and dead seems to be needed to stain which part.

We thank the Reviewer for this suggestion. In this amended version of the manuscript, we added this information in the figure legend. Specifically, the cell-permeant dye Calcein AM was used as a live cell indicator and BOBO-3 Iodide was used as a dead cell indicator.

  1. Figure 6C. It seems necessary to additionally present an experiment to compare and confirm the 3D condition and the 3D+CAF condition.

We thank the Reviewer for this comment. As stated above, in this amended version of the manuscript, we compared the 3D condition and the 3D+CAF condition by quantifying tumorsphere size and roundness and presented these data as bar graphs in new Supplementary Figure 2a (left and right panel).

  1. IRB number missing, it seems to need correction.

We thank the Reviewer for this suggestion. In this amended version of the manuscript, we added the following information in the Materials and Methods section: Ethics Committee name: Comitato Etico Istituto tumori "Giovanni Paolo II", Istituto di ricovero e cura a carattere scientifico, viale Orazio Flacco, 65-70124, Bari. Approval Code: Prot. n. 379/C.E. Approval Date: 16/09/2020.

Review report (round 2)

Although the authors appear to have solved many existing problems through their review, it is confirmed that there are still many problems. It seems that the following problems need to be resolved.

We appreciate that the reviewer recognized our work since we responded to all of its remarks raised during first round of revision.

  1. Although the title says that SMYD3 regulates HGF-MET signaling, this paper mainly focuses on inhibition of SMYD3. In order to show the mechanism of HGF-MET signal transduction by SMYD3, it is necessary to prove the mechanism of MET by SMYD3.

We focused our attention on investigating the role of SMYD3 in MET signaling pathway in gastric cancer, for this reason we examined in this manuscript the pharmacological inhibition of SMYD3. The point raised by the reviewer, in this second round of revision, could be further explore in a future paper.   

  1. It is believed that the p-MET data shown in Figure 2a should be shown in one gel for concentration experiments of 50ng, 150ng and 300ng so that comparison of concentrations can be made clearer.
  2. Figures 3C and 4C basically show that there is phosphorylation of MET, but looking at the data in Figure 2a, it appears that phosphorylation of MET is not high. I think a comparative explanation as to why they are different is needed. Did you process the inhibitor after 300ng of HGF?

Both of these points were not raised during the first round of revision by reviewer 2.

  1. I think the data in figure 6C may also be needed in figure 5c. Additionally, if you want to see cleaved caspase3 and PARP, it seems necessary to show the total form as well for comparison.

We added the figure 6c in the amended version of our manuscript in order to respond to reviewer 3’s remark, for this reason it should not be discussed in a second round of revision.

  1. I think a subtitle for supplementary figures 1 and 2 is needed.
  2. Although many corrections have been made, typos still exist.
  3. ex) figure S21 -> figure S2

We rechecked our manuscript searching for typos but we didn’t find them. Also the example indicated by the reviewer was actually correctly indicated in the amended version of manuscript, but it was showed in tracking change modality (as request by “Cells” guidelines) that’s why it can have been missed by the reviewer.  
